# Lookers-on see most of the game: External insight-guided method for enhancing uncertainty estimation

## Abstract

Large Language Models (LLMs) have gained increasing attention for their impressive capabilities, alongside concerns about the reliability arising from their potential to generate hallucinations and factual inaccuracies. Uncertainty estimation for LLMs aims to quantify the uncertainty of model outputs, where high uncertainty scores indicate potential errors, signaling the need for rejection or further evaluation. However, existing methods often limited by inherent biases of LLMs like over-confidence and under-confidence. In this paper, we propose an external insight-driven correction method for refining uncertainty estimation. This method integrates uncertainty scores derived from a lightweight model trained on global information with those from existing uncertainty estimation approaches, providing a more robust solution. We present comprehensive experimental results that demonstrate the effectiveness and generalizability of our method across various models, datasets, and consistently surpassing all baselines.

## 1 Introduction

Large Language Models (LLMs) have demonstrated their impressive capability in natural language understanding and generation (Karanikolas et al., 2023), so as to provide valuable assistance in numerous applications such as free-form question answering (Joshi et al., 2017) and decision-making (Li et al., 2023). However, challenges related to output reliability, including hallucinations (Zhang et al., 2023) and factual inaccuracies (Wachter et al., 2024), which mislead users with false information, remain a significant concern. These issues are especially critical in high-risk applications like medical diagnosis (Wang et al., 2023) and legal consultation (Cheong et al., 2024), making it essential to accurately assess the reliability of LLM outputs.

Uncertainty estimation (Loquercio et al., 2020) is a critical component in ensuring the reliability of LLMs in practical applications. A reliable estimation of uncertainty can help to determine when to trust a model (Yadkori et al., 2024). Intuitively, we would expect a high uncertainty estimation when the response of model is likely to be incorrect, which should either be rejected or further evaluated.

Prior research (Papadopoulos & Yeung, 2001; Gal & Ghahramani, 2016) on uncertainty estimation has primarily concentrated on scenarios such as classification or regression using targeted uncertainty techniques, which are not directly applicable to LLMs. Consequently, there has been growing interest in developing uncertainty estimation methods tailored for LLMs, which can be broadly categorized into logit-based methods (Malinin & Gales, 2020; Kuhn et al., 2023), verbalized methods (Lin et al., 2023; Xiong et al., 2023), internal state-based methods (Kadavath et al., 2022; Ji et al., 2024) and consistency-based methods (Li et al., 2024b; Pedapati et al., 2024). While these methods provide valuable insights, their reliance on model outputs—whether at the logits level or text level—often confines uncertainty estimation to inherent biases of LLMs, especially the over-confidence and under-confidence (Ye et al., 2024), as illustrated in Table 1.

Over-confidence is a common issue in LLMs, characterized by their tendency to hallucinate facts and present inaccuracies in a confident manner when composing responses (Azaria & Mitchell, 2023). From a deeper perspective, it involves assigning high-probability values to incorrect responses (Yadkori et al., 2024). From a broader viewpoint, it exhibits consistent incorrect response across multiple samplings. For example, when queried about "What movie was about a dunking

Table 1: Instances of over-confidence and under-confidence in the LLaMA-3-8B-Instruct model. For the first question, the model repeatedly provided the same but incorrect responses in multiple samplings, illustrating explicit over-confidence. In contrast, for the second question, the model generated varied responses, with the top response being correct, showcasing explicit under-confidence.

| Query | Answer | Response Samples | Top Response |
|---|---|---|---|
| What movie was about a dunking Golden Retriever named Buddy? | Air Bud | Elf ✗
Elf ✗
Elf ✗ | Elf ✗ |
| Who was born at Villa Mon Repos, Corfu, in 1921? | Prince Philip | Queen Elizabeth II ✗
King Paul ✗
King Constantine ✗ | Prince Philip ✓ |

Golden Retriever named Buddy?", a question sampled from TriviaQA (Joshi et al., 2017), the target model consistently produces "Elf" in every separate sampling, whereas the real answer is "Air Bud". This consistent incorrection highlights a divergence between the model's internal consistency and the external accuracy, which can adversely affect uncertainty estimation, leading to relatively low uncertainty score despite the presence of inaccuracy. Conversely, under-confidence refers to the opposite scenario, where the target model is capable of correctly responding a question but is perceived as having high uncertainty due to the diverse outputs generated during sampling. Both phenomena negatively impact uncertainty estimation performance, primarily leading to the relative inversion of uncertainty scores ranking between questions the model can and cannot answer correctly.

Consequently, it is imperative to focus on the rank ordering of the uncertainty scores, which is also known as relative uncertainty scores. Given the inherent misalignment between the model confidence and its actual knowledge ability, along with the impracticality of modifying the model directly, a heuristic approach is correcting the inversion of rank ordering derived from model-dependent methods by integrating correction scores. Accordingly, we propose an external insights-driven method to augment uncertainty estimation, orthogonal to existing advanced methods such as Semantic Entropy (SE) (Kuhn et al., 2023) and Shifting Attention to Relevance (SAR) (Duan et al., 2023), both of which can be easily integrated with our method.

Specifically, we begin by meticulously curating supervised dataset that are closely aligned with the target LLM's performance within a particular domain of knowledge. This dataset is then used to train an auxiliary lightweight model, which serves as a *Corrector*. By integrating the *Corrector* trained on global information with those uncertainty estimation methods that rely solely on the target model, we can significantly refine the uncertainty scores.

Our main contributions are thus as follows:

- We identify the limitations of current uncertainty estimation methods which suffer from inherent biases in LLMs, including over-confidence and under-confidence. Additionally, we provide both theoretical proof and empirical evidence.

- We propose an external insight-driven approach which enables seamless integration with existing uncertainty estimation methods. This approach can correct the inversion of uncertainty score rankings caused by the inherent biases of LLMs.

- We demonstrate that our method consistently outperforms existing approaches included in the Representative Baselines Set (RBS) and the Challenging Baselines Set (CBS), exhibiting significant improvements in both relative and absolute terms. Furthermore, we present comprehensive experimental evidence underscoring the robustness and generalizability of our approach across diverse data domains and target models.

## 2 PRELIMINARIES

In this section, we commence by clarifying the two scales of uncertainty: *relative uncertainty* and *absolute uncertainty*. We then formalize the relative uncertainty estimation as a classification task to determine whether the target model can correctly respond to a given question. Subsequently, we

delve into the theoretical foundations of widely-used logit-based uncertainty estimation methods, and critically examine the inherent limitations shared by those approaches that rely exclusively on target model outputs.

## 2.1 RELATIVE UNCERTAINTY AND ABSOLUTE UNCERTAINTY

Research on uncertainty estimation has led to two key concepts (Kamath et al., 2020; Vazhentsev et al., 2023): *relative uncertainty* and *absolute uncertainty*, each providing distinct methods for assessing and interpreting levels of uncertainty. Given an input $x$, a ground truth answer $y$, and the predictive distribution of $Y$, the predictive uncertainty for the target model regarding the input $x$ is denoted as $\mathrm{UE}(x, \theta)$. Relative uncertainty scores emphasize the accuracy of sample ranking, especially in discerning questions that the target model can correctly respond to from those it struggles with. Ideally, for every pair $(x_i, y_i)$ and $(x_j, y_j)$ with their predictive distributions $Y_i$ and $Y_j$, we should have

$$\mathrm{UE}(x_i, \theta) \leq \mathrm{UE}(x_j, \theta) \iff P(Y_i = y_i | x_i, \theta) \geq P(Y_j = y_j | x_j, \theta). \tag{1}$$

Stricter than relative uncertainty scores, absolute uncertainty scores support to represent the model's accuracy. In cases where there is an 80% uncertainty prediction, it implies that the question is expected to be answered correctly only 20% of the time under similar conditions. This relationship can be mathematically expressed as

$$P(Y = y | \mathrm{UE}(x, \theta) = q) = 1 - q. \tag{2}$$

As relative uncertainty concerns solely with the relative rankings of $h(x) = \mathrm{UE}(x, \theta)$, it can be framed as a classification problem aimed at finding a function $h$ that minimizes the expected loss of misclassification (Allikivi et al., 2024; Tao et al., 2023). Consider two class labels, $\mathcal{C} = \{c_0, c_1\}$, indicating whether the targrt model can correctly answer the question or not, respectively. This leads to the formulation of a decision rule

$$g(h; \tau) = \begin{cases} c_0 & \text{if } h(x) \leq \tau \text{ (confident)} \\ c_1 & \text{if } h(x) > \tau \text{ (uncertain)} \end{cases}, \tag{3}$$

where $h(x)$ is a scalar measure of uncertainty and $\tau$ is the threshold.

Drawing from decision theory, we derive the expected loss as *conditional risk* for the sample $x$:

$$\mathrm{Risk}(x) = \lambda_{c_i, c_{1-i}} h_{c_{1-i}}(x), \tag{4}$$

where $c_i, i \in \{0, 1\}$ denotes the true label of the sample $x$, and $h_{c_{1-i}}(x) = P(c_{1-i} \mid x)$ is the posterior probability of misclassifying the sample $x$ as class $c_{1-i}$. $\lambda_{c_i, c_{1-i}}$ represents the loss associated with this misclassification—specifically, a penalty incurred when the sample with the label $c_i$ is classified as $c_{1-i}$. Our task is to find $h^*$ that minimizes the overall risk

$$\mathrm{Risk}(h) = \mathbb{E}_x \left[ \mathrm{Risk}(h(x)) \mid x \right]. \tag{5}$$

## 2.2 THEORETICAL FOUNDATIONS OF UNCERTAINTY ESTIMATION FOR LLM

LLMs typically generate outputs in an auto-regressive manner, which iteratively predict the probability distribution of the subsequent token based on the evolving context (Gregor et al., 2014). Given an input sequence $x$ with the objective of generating an output sequence $y = \{y_1, y_2, \ldots, y_L\}$, the conditional probability of the $l$-th token $y_l$ is denoted as $P(y_l | y_{<l}, x; \theta)$. This probability depends on all previously generated tokens $y_{<l} = \{y_1, y_2, \ldots, y_{l-1}\}$ as well as the input $x$. The probability of generating the entire sequence $y$ can be expressed as the product of the conditional probabilities of each individual token:

$$P(y|x; \theta) = \prod_{l=1}^{L} P(y_l | y_{<l}, x; \theta), \tag{6}$$

where $P(y_l | y_{<l}, x; \theta) = \frac{e^{z_l / T}}{\sum_j e^{z_j / T}}$, $z$ is the raw logit, and $T$ is the temperature that controls the smoothness of the probability distribution. This posterior probability provides a probabilistic framework for sequence generation. Moreover, according to prior research (Malinin & Gales, 2020), the total uncertainty for the generation of $y$ is given by the entropy of the predictive posterior:

$$PE(x) = \mathcal{H}[P(y \mid x, \theta)] = \mathbb{E}_{P(y|x, \theta)}[-\ln P(y \mid x, \theta)] = -\sum_{y \in Y} P(y \mid x, \theta) \ln P(y \mid x, \theta). \tag{7}$$

In practice, due to the exponential computational complexity of traversing the entire response set, Monte Carlo approximation method (Papadopoulos & Yeung, 2001) is employed via beam search with a single target model for generation. The approximate entropy is defined as

$$PE(x) \approx -\frac{1}{B} \sum_{b=1}^{B} \ln P(y_b|x,\theta), \tag{8}$$

where $P(y_b|x,\theta)$ denotes the posterior probability of the $b$-th beam search candidate. Base on these, Kuhn et al. (2023) proposed to cluster generations with similar meanings and compute entropy using the probabilities associated with each semantic cluster. This approach is formulated as

$$SE(x,\theta) = -\frac{1}{C} \sum_{i=1}^{C} \ln P(c_i|x,\theta), \tag{9}$$

where $c_i$ denotes each semantic cluster and $C$ represents the set of all clusters.

Another form of improvement is to assign weights to each token in the generation when calculating posterior probabilities (Duan et al., 2023; Bakman et al., 2024), either through a manually designed algorithm or a training way, which can be formulated as

$$\tilde{P}(y \mid x; \theta) = \prod_{l=1}^{L} P(y_l \mid y_{<l}, x; \theta) \cdot w_l, \tag{10}$$

where $w_l$ represents the weight assigned to the $l$-th token.

## 2.3 CHALLENGES IN UNCERTAINTY ESTIMATION

Our previous discussions have centered on logits-based methods, recognized for their widespread use, effectiveness, and solid theoretical foundation. In contrast, other types of methods like verbalized methods and internal state-based methods often lack stable theoretical frameworks, with empirical evidence in Tables 2 and Tables 3 showing their performance generally falling short of advanced logits-based methods. Therefore, we focus on logit-based methods and demonstrate their inherent limitations in the context of LLMs.

Over-confidence and under-confidence represent persistent challenges in LLMs.

Over-confidence can be expressed when the probability of a specific incorrect response $P(y' \mid x; \theta)$ significantly exceeds the total probability of all other possible responses, including the correct answer $P(y^* \mid x; \theta)$. This can be represented as

$$P(y' \mid x; \theta) \gg \sum_{y_i \in Y, y_i \neq y'} P(y_i \mid x; \theta), \tag{11}$$

where $Y$ represents the set of all possible responses. As a result, during sampling, the specific incorrect response tends to be generated with high probability, overshadowing other responses.

Conversely, under-confidence occurs when

$$P(y^* \mid x; \theta) = \max_{y} P(y \mid x; \theta) \quad \text{and} \quad \exists S \subseteq Y, \forall y_j \in S, \ P(y^* \mid x; \theta) - P(y_j \mid x; \theta) < \delta, \tag{12}$$

where $\delta$ represents a small positive number. This suggests that while the correct response $y^*$ has the highest probability of being sampled, there are also incorrect responses that compete closely with it.

The over-confidence and under-confidence inherent in LLMs can significantly influence uncertainty estimation by transferring to logit-based methods through their anomalous confidence expressions.

In cases of over-confidence, the probability of a specific incorrect response $y'$ converges to a value $p'$, where $p' \rightarrow 1$. This scenario can be denoted as $P(y' \mid x; \theta) = p'$ and $P(y_i \mid x; \theta) = \epsilon$ for all $y_i \neq y'$, with $\epsilon$ being sufficiently small. Under these conditions, the uncertainty $U(x)$ can be expressed as

$$U(x) = \mathcal{H}[P(y \mid x, \theta)] \approx -p' \ln p' - \sum \epsilon \ln \epsilon. \tag{13}$$

As $p' \rightarrow 1$ and $\epsilon \rightarrow 0$, it follows that $-p' \ln p' \rightarrow 0$ and $-\epsilon \ln \epsilon \rightarrow 0$. Consequently, the overall entropy approaches 0, resulting in a substantial low number of the uncertainty score.

In under-confidence, we assume $P(y^* \mid x; \theta) = p^*$ and $P(y_i \mid x; \theta) = p_i$ for $y_i \neq y^*$, where $p^* + \sum_{y_i \neq y^*} p_i = 1$. By Jensen's inequality, we have:

$$- \sum_{y_i \neq y^*} p_i \ln p_i > -(1 - p^*) \ln(1 - p^*), \tag{14}$$

which indicates that the contribution from $-\sum_{y_i \neq y^*} p_i \ln p_i$ is considerable. Since there exist $p_j$ values comparable to $p^*$, under the conditions specified in 12, as $p^*$ decreases, $1 - p^*$ increases, leading to higher computed entropy.

Consequently, we express $U(x)$ as

$$U(x) = -p^* \ln p^* - \sum_{y_i \neq y^*} p_i \ln p_i > -p^* \ln p^* - (1-p^*) \ln(1-p^*) > -2(1-p^*) \ln(1-p^*). \tag{15}$$

Based on the previous discussion, we can conclude that $U(x)$ is overestimated due to the considerable contribution of $1 - p^*$.

## 3 METHODOLOGY

In this section, we introduce an external insight-driven method to refine uncertainty estimation, which integrates uncertainty scores derived from a lightweight model trained on global information with those from existing uncertainty estimation approaches. Through this method, we provide a more robust solution for uncertainty estimation, effectively mitigating the adverse effects of inherent biases in LLMs.

Our method comprises three main steps including *dataset crafting, corrector training and uncertainty correcting*. Firstly, we carefully construct a dataset that closely aligns with the target model's performance within a particular domain of knowledge. This dataset is then utilized to train an auxiliary lightweight model that serves as a correction module, facilitating seamless integration with existing uncertainty estimation methods to obtain corrected uncertainty scores.

**Step 1: Dataset Crafting**

We start by extracting data from existing datasets to serve as a evaluation set for assessing the model's capabilities on a particular domain of knowledge. This evaluation set comprises a collection of question-answer pairs, denoted as $\mathcal{D} = \{(q_i, a_i) \mid i = 1, \ldots, n\}$. For each question $q_i$, we engage the target model $M$ to generate a corresponding response $r_i$, thereby obtaining the response set $\mathcal{R} = \{r_i \mid i = 1, \ldots, n\}$. Afterward, we evaluate each model response $r_i$ against its corresponding ground truth answer $a_i$, employing both rule-based and LLM-based methods to ensure an accurate assessment. A binary label $c_i$ is then assigned to each sample, defined as

$$c_i = \begin{cases} 1 & \text{if } r_i \text{ is equivalent to } a_i \\ 0 & \text{otherwise} \end{cases} \tag{16}$$

By pairing question $q_i$ with its binary label $c_i$, we form a correction dataset $\mathcal{D}_{\text{cor}} = \{(q_i, c_i) \mid i = 1, \ldots, n\}$. This dataset provides external insight into target model's performance in generating correct responses. To directly associate the questions with uncertainty, we transform the form of dataset into $\mathcal{D}^*_{\text{cor}} = \{(q_i, 1 - c_i) \mid i = 1, \ldots, n\}$.

**Step 2: Corrector Training**

Following the discussion in 2.1, we frame uncertainty estimation as a classification task, focusing on the relative uncertainty score rankings between questions the model can answer correctly and those it cannot, thereby defining two distinct classes. Thus, we train a classifier using the curated dataset $\mathcal{D}^*_{\text{cor}}$ to determine whether the target model **fail** to correctly answer a given question. The classifier integrates a fully connected layer following the RoBERTa model (Liu, 2019), with the representation of the $[CLS]$ token as its input, denoted as $\mathbf{h}_{[CLS]} \in \mathbb{R}^d$. After applying a sigmoid activation function $\sigma(z)$, we get the output value $\hat{y}_i = \sigma(\mathbf{W} \cdot \mathbf{h}_{[CLS]} + b)$, which falls within the range $[0, 1]$. After training, we develop a *Corrector* that effectively aligns its output scores with the target model's performance. Same as traditional uncertainty scores, a higher output value signifies a state of high uncertainty.

**Step 3: Uncertainty Correcting**

In this step, we integrate uncertainty score $U(x)$ derived from other uncertainty estimation methods based on model itself with the correction score $C(x)$ computed by *Corrector*. Specifically, we employ a weighted combination method to integrate the two scores and apply grid search to systematically evaluate the hyperparameter $w$. With the optimal weight $w^*$ searched in development dataset, the integrated uncertainty score $U_{\mathrm{cor}}(x)$ can be expressed as

$$U_{\mathrm{cor}}(x) = w^* \cdot U(x) + (1 - w^*) \cdot C(x) \tag{17}$$

## 4 EMPIRICAL EVALUATION

In this section, we demonstrate that our *Corrector* is a effective robust module for enhancing the performance of uncertainty estimation in LLMs.

### 4.1 EXPERIMENTAL SETUP

**Target Model** Since model size is not the primary focus of our investigation, we select OPT-2.7B (Zhang et al., 2022), which is widely used in prior works (Kuhn et al., 2023; Duan et al., 2023) as target model. We also considered the advanced open-source model LLaMA-3-8B-Instruct (Dubey et al., 2024) as target model for the main experiments.

**Metrics** Following prior works (Kuhn et al., 2023; Duan et al., 2023), we use the area under the receiver operating characteristic curve (**AUROC**) as our primary metric for uncertainty estimation, which is a commonly used metric for classification tasks. In our experiments, it can be used to evaluate the performance of *relative uncertainty*. AUROC of 1 indicates that the uncertainty estimation method perfectly differentiates between questions the target model can respond correctly and those it cannot, whereas an AUROC of 0.5 suggests that the estimation is no better than random guessing. Expected Calibration Error (**ECE**) is another metric we use, which can evaluate the performance of absolute uncertainty. In ECE, confidence and uncertainty are treated as complementary values, with the confidence score being computed as 1 minus uncertainty score. ECE is calculated by partitioning predicted confidence scores into bins and comparing the average confidence in each bin to the actual fraction of correct predictions, formalized as

$$\mathrm{ECE} = \sum_{m=1}^{M} \frac{|B_m|}{n} \left| \mathrm{acc}(B_m) - \mathrm{conf}(B_m) \right|, \tag{18}$$

where $M$ is the number of bins, $B_m$ is the set of predictions in bin $m$, $|B_m|$ is the number of samples in that bin, and $n$ is the total number of samples.

**Datasets** We focus on the question-answering task using two representative datasets: **TriviaQA** (Joshi et al., 2017) and **SciQA** (Auer et al., 2023). **TriviaQA** comprises 95,000 question-answer pairs created by trivia enthusiasts, supplemented with independently sourced evidence documents. We utilize TriviaQA as a closed-book task, where target models are challenged to provide answers without access to supporting paragraphs. **SciQA** contains 2,565 question-answer pairs fetched from the open research knowledge graph, covering several research fields ranging from science and technology like Computer Science, Engineering, Chemistry, and Geology, life sciences like Immunology and Genetics to social sciences like Economics and Urban Studies.

**Baselines** We select various uncertainty estimation methods as baselines and we divide them into two sets: **Representative Baseline Set (RBS)** to evaluate the general applicability of our method across various categories of representative approaches, and **Challenging Baseline Set (CBS)** to assess its effectiveness against more challenging approaches. **RBS** includes representative uncertainty estimation methods from logit-based, verbalized, internal state-based, and consistency-based categories, specifically: Lexical Similarity (**LS**) (Fomicheva et al., 2020) computing similarities among multiple sentences as a measure of consistency, Verbal Confidence (**VC**) (Xiong et al., 2023) requiring the target model to respond and provide confidence score, **P(True)** (Kadavath et al., 2022) first asking the target model to propose an answer and then execute self-evaluation in a internal probability way, and Predictive Entropy (**PE**) (Malinin & Gales, 2020) computing uncertainty using the entropy of the predictive posterior. In addition to being representative, these baselines can be easily implemented. As for the **CBS**, we delve into a series of logits-based methods that demonstrate superior performance, including Length-normalized Predictive Entropy (**LN-PE**) (Malinin

| Method | TriviaQA | | | | | | SciQA | | | | | |
|---|---|---|---|---|---|---|---|---|---|---|---|---|
| | AUROC(↑) | | | ECE(↓) | | | AUROC(↑) | | | ECE(↓) | | |
| | Vanilla | +Corrector | Improv | Vanilla | +Corrector | Improv | Vanilla | +Corrector | Improv | Vanilla | +Corrector | Improv |
| | | | | | | OPT-2.7B | | | | | | |
| LS | 42.30 | **70.01** | +27.71 | 78.51 | **17.30** | -61.21 | 53.02 | **63.33** | +10.31 | 70.00 | **32.78** | -37.22 |
| VC | 44.50 | **71.50** | +27.00 | 75.00 | **20.03** | -54.97 | 48.34 | **55.63** | +7.29 | 68.40 | **35.04** | -33.36 |
| P(True) | 49.00 | **72.75** | +23.75 | 63.32 | **18.50** | -44.82 | 51.54 | **60.60** | +9.06 | 66.34 | **34.52** | -31.82 |
| PE | 47.69 | **69.98** | +22.28 | 50.22 | **17.47** | -32.75 | 50.40 | **62.65** | +12.25 | 62.13 | **36.92** | -25.21 |
| | | | | | | LLaMA-3-8B-Instruct | | | | | | |
| LS | 19.57 | **69.82** | +50.25 | 70.25 | **7.41** | -62.84 | 53.67 | **65.38** | +11.71 | 38.64 | **18.19** | -20.45 |
| VC | 62.34 | **74.89** | +12.55 | 23.41 | **16.78** | -6.63 | 68.22 | **72.15** | +3.93 | 31.88 | **19.47** | -12.36 |
| P(True) | 57.14 | **72.29** | +15.15 | 24.67 | **19.84** | -4.83 | 65.63 | **71.41** | +5.78 | 34.56 | **31.92** | -2.64 |
| PE | 64.52 | **69.76** | +5.25 | 21.38 | **17.24** | -4.13 | 66.54 | **67.98** | +1.44 | 40.67 | **34.07** | -6.60 |

Table 2: AUROC and ECE scores on the TriviaQA and SciQA datasets obtained by applying our method to baselines from the **Representative Baseline Set (RBS)**. LS, VC, and PE denote the Lexical Similarity method, Verbal Confidence, and Predictive Entropy, respectively.

| Method | TriviaQA | | | | | | SciQA | | | | | |
|---|---|---|---|---|---|---|---|---|---|---|---|---|
| | AUROC(↑) | | | ECE(↓) | | | AUROC(↑) | | | ECE(↓) | | |
| | Vanilla | +Corrector | Improv | Vanilla | +Corrector | Improv | Vanilla | +Corrector | Improv | Vanilla | +Corrector | Improv |
| | | | | | | OPT-2.7B | | | | | | |
| LN-PE | 52.27 | **70.96** | +18.69 | 45.66 | **31.59** | -14.07 | 44.82 | **62.16** | +17.34 | 32.74 | **34.68** | -15.19 |
| SE | 65.57 | **73.88** | +8.31 | 44.78 | **31.54** | -13.24 | 57.60 | **59.67** | +2.07 | 52.67 | **42.23** | -10.44 |
| SAR-t | 58.50 | **72.14** | +13.64 | 41.47 | **29.05** | -12.42 | 57.21 | **63.92** | +6.71 | 52.18 | **44.19** | -7.99 |
| SAR-s | 49.29 | **69.90** | +20.61 | 71.66 | **26.25** | -45.41 | 50.98 | **62.01** | +11.03 | 51.48 | **34.83** | -16.65 |
| SAR | 57.04 | **71.32** | +14.28 | 40.50 | **28.38** | -12.12 | 58.40 | **64.97** | +6.57 | 43.18 | **38.99** | -4.19 |
| | | | | | | LLaMA-3-8B-Instruct | | | | | | |
| LN-PE | 72.55 | **74.79** | +2.24 | 14.31 | **11.53** | -2.79 | 69.48 | **71.56** | +2.08 | 29.38 | **23.76** | -5.62 |
| SE | 80.92 | **82.12** | +1.20 | 13.07 | **12.76** | -0.31 | 71.59 | **72.93** | +1.34 | 30.54 | **25.23** | -5.30 |
| SAR-t | 79.55 | **79.93** | +0.38 | 16.40 | **13.70** | -2.70 | 72.26 | **73.87** | +1.61 | 30.37 | **26.81** | -3.56 |
| SAR-s | 69.87 | **77.09** | +2.95 | 6.73 | **20.00** | -3.17 | 74.96 | **75.72** | +0.76 | 38.54 | **36.18** | -2.37 |
| SAR | 80.92 | **81.90** | +0.98 | 16.17 | **13.76** | -2.41 | 73.88 | **75.19** | +1.31 | 28.97 | **25.60** | -3.37 |

Table 3: AUROC and ECE scores on the TriviaQA and SciQA datasets obtained by applying our method to baselines from the **Challenging Baseline Set (CBS)**. LN-PE, SE, denote the Length-normalized Predictive Entropy method, and Semantic Entropy, and Predictive Entropy, respectively. SAR-t refers to the token-level version of the SAR method, while SAR-s denotes the sentence-level version.

& Gales, 2020), which adjusts PE by normalizing it based on sentence length, Semantic Entropy (**SE**) (Kuhn et al., 2023), which clusters sentences with equivalent meanings and calculating cluster-wise entropy, Shifting Attention to Relevance (**SAR**) (Duan et al., 2023), which encompasses both token-level shifting (**SAR-t**) and sentence-level shifting (**SAR-s**).

**Evaluation of Model Responses** We evaluate model responses using a combination of fuzzy matching and LLM evaluation. Compare to prior work (Kuhn et al., 2023), we employ fuzzy matching with an increased acceptance threshold for correct answers: $\mathcal{M}(y, y') = \mathbb{I}_{RougeL(y,y')>0.5}$. A response $y$ is considered correct only if its longest common subsequence score exceeds 0.5 compared to the reference answer $y'$. Additionally, we use GPT-turbo-3.5-0613 (Ouyang et al., 2022) to compare the model response with the reference answer and assess its correctness. The combination of these two methods enhances the precision of our evaluation.

## 4.2 RESULTS & ANALYSIS
### 4.2.1 COMPARISON WITH BASELINES

We compare the results of our method with vanilla baselines from the **RBS** and **CBS** in Table 2 and Table 3. It is illustrated that our method consistently improves the performance of all baselines across various datasets and target LLMs. Given the numerous baselines, we simplify our expressions by using the average score of AUROC within each baselines set, and the discussion regarding ECE will be reserved for Section4.2.3.

**RBS** In Table 2, we focus on methods that belong to different categories as summarized in Section 5. For each of the four categories—consistency-based methods, verbal confidence methods, internal state-based methods, and logit-based methods—we have selected one representative method, denoted as LS, VC, P(true), and PE, respectively. As illustrated in Table 2, the AUROC scores for each baseline across the target models exhibit extremely low performance, especially in the Trivi-aQA dataset, with an average AUROC score of 0.48, which is even worse than random guessing.

|  | TriviaQA | SciQA |
|---|---|---|
| TriviaQA | 19.59 | 4.05 |
| SciQA | 6.03 | 10.20 |

(a) Generalization for Domain of Data

|  | OPT-2.7B | OPT-6.7B | LLaMA3-8B |
|---|---|---|---|
| OPT-2.7B | 19.59 | 11.80 | 3.23 |
| OPT-6.7B | 6.08 | 11.21 | 3.43 |

(b) Generalization for Target Model

Table 4: Average AUROC scores improvement of after appling our method to baselines. (a) The leftmost column indicates the domains of data used in training, while the topmost row represents the domains of data used for evaluating, with OPT-2.7B serving as the target model. (b) The leftmost column denotes the target model during training, whereas the topmost row signifies the target model during evaluating, with TriviaQA utilized as the target domain of data.

If we only consider OPT-2.7B, the score further decreases to 0.46. These poor performances indicate a trade-off between ease of implementation and robustness for uncertainty estimation methods, revealing the significant optimization potential inherent in these baselines. Notably, the application of our *Corrector* to the representative baselines yields average AUROC scores of 0.71 and 0.63 on TriviaQA and SciQA, respectively. These results reflect significant improvements of 0.27 and 0.09, even exceeding the performance of challenging baseline such as SAR.

**CBS** We select a series of logits-based methods with strong performance to test our *Corrector* against challenging baselines. These challenging baselines mainly anchor their estimation on the target model's outputs—at eithor the logits level or text level—and make tailored adjustments to the predictive entropy framework based on issues observed in natural language generation tasks of LLMs. As illustrated in Figure 1, while challenging baselines show significant improvements over PE and other representative baselines, narrowing the gap to the ground truth, there remains a considerable disparity. This provides room for our *Corrector*. The results presented in Table 3 demonstrate that incorporating the *Corrector* yields average AUROC score of 0.75 for TriviaQA and 0.63 for SciQA, with improvements of 0.08 and 0.05, respectively. Furthermore, peak AUROC scores of 0.82 and 0.75 can be reached.

Figure 1: Density plot of uncertainty scores on TriviaQA with OPT-2.7B as the target model, obtained from various baselines and ground-true.

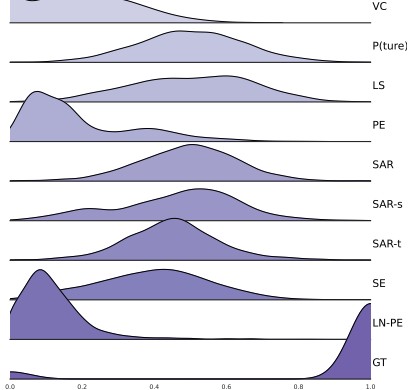

**Target models** In the preceding discussion, we did not differentiate among various target models. When using the earlier model OPT-2.7B as the target model, both the representative baselines and the challenging baselines yielded poor results, especially on the SciQA dataset. Although applying our Corrector resulted in some improvement, there remains room for further enhancement compared to our overall results. In contrast, when using the advanced model LLaMA-3-8B-Instruct as the target model, the baselines outperform those based on OPT-2.7B, with substantial improvements ranging from 0.08 to 0.20. This indicates that enhancement in LLM capabilities may significantly improve the confidence estimation performance.

### 4.2.2 GENERALIZATION

The above results indicate that the *Corrector* performs effectively on the evaluation set comprising in-distribution data. However, our analysis highlights two primary variables that can lead to out-of-distribution scenarios: **domain of data** and **target model**. The generalization performance of the *Corrector* is evaluated through the average improvement of AUROC scores across all baselines from both RBS and CBS.

**Domain of Data** To evaluate the generalization capability of our *Corrector* across different data domains, we conduct experiments by training the *Corrector* on the dataset $\mathcal{D}^*_{cor}$, crafted from either TriviaQA or SciQA, and then evaluating it on the alternate one. As illustrated in Table 4, the *Corrector* achieves optimal performance when both training and evaluating occur within the same data domain. Remarkably, even when training and evaluating on different domains, the *Corrector* still demonstrates a enhancement, yielding an average improvement of approximately 0.05. One possibility is that the target model exhibits comparable knowledge proficiency across both data domains.

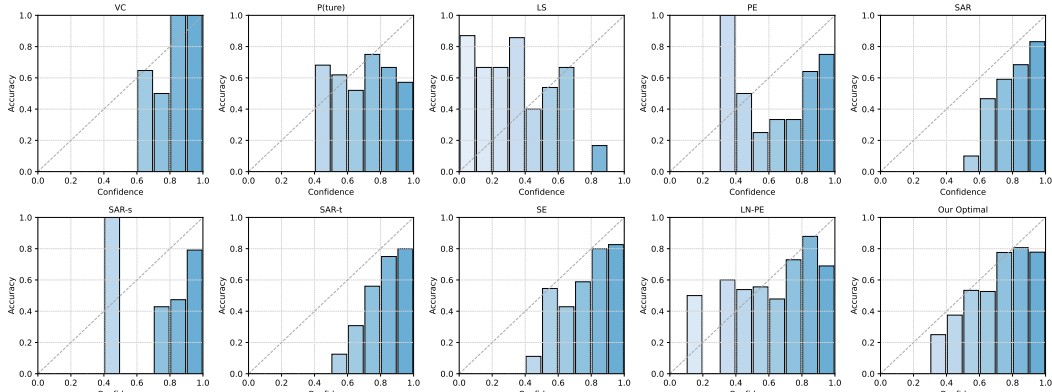

Figure 2: Calibration Plot. This plot illustrates the relationship between predicted confidence and observed frequencies. The diagonal line represents perfect calibration, indicating that the predicted confidence matches the actual outcomes. The bars that extend above the diagonal line indicate an underestimation of confidence, while those that fall below the diagonal line signify an overestimation of confidence.The last plot illustrates the optimal performance achieved by applying our method.

**Target Model** We investigate the generalization for target model by training *Corrector* on $\mathcal{D}^*_{cor}$ sourced from a different target model than the one used for evaluating. As shown in Table 4, in cases where models exhibit relatively comparable knowledge capabilities, such as OPT-2.7B and OPT-6.7B, the *Correcter* exhibits generalization ability, yielding average AUROC improvements of 0.11 and 0.06, respectively. Conversely, when a substantial performance gap exists between models, such as OPT-2.7B and LLaMA-3-8B-Instruct, we achieve an average AUROC improvement of 0.03. When focusing solely on the challenging baselines from CBS, the improvement drops to 0.01.

### 4.2.3 CALIBRATION

In our discussion on calibration, we adhere to the experimental settings as previously outlined in Section 4.2.1. As detailed in Section 4.1, we treat confidence and uncertainty as two intrinsically connected facets to compute ECE. In a group of questions with an uncertainty score of 0.8, corresponding to a confidence level of 0.2, it can be inferred that there is a 20% likelihood that this group of questions will be answered correctly by target model . As shown in Table 2, most prior methods have overlooked the calibration issue, leading to relatively poor performance on ECE scores, which reflect inadequate absolute uncertainty. Although calibration is not the primary focus of our method, experimental results demonstrate that our approach significantly enhances calibration performance, yielding substantial lower ECE scores compared to all baselines.

For instance, when using the OPT-2.7B as the target model, we observed an average reduction of 0.34 and 0.21 on TriviaQA and SciQA respectively, across all baselines. With the LLaMA-3-8B-Instruct model as the target, the reductions are 0.11 and 0.07 respectively, which are still considerable. This significant reduction underscores the remarkable efficacy of our method in enhancing uncertainty estimation in a absolute way. Additionally, we employ calibration plots, as depicted in Figure 2, to visually demonstrate the calibration effectiveness of our method.

## 5 RELATED WORK

Uncertainty estimation methods for LLMs have gained significant attention, with approaches can be broadly categorized into logit-based methods, verbal confidence, internal state-based methods, and consistency-based approaches.

**Logit-based methods** Logit-based methods are the most widely used and effective approaches in uncertainty estimation. As a foundational method, Predictive Entropy (PE) (Malinin & Gales, 2020), defines total uncertainty as the entropy of the output logits distribution. Follow that, Kuhn et al. (2023) introduced semantic entropy (SE) that estimates uncertainty by marginalizing over semantically-equivalent samples in NLG tasks. In the similar framework, Nikitin et al. (2024) employed positive semi-definite kernels and von Neumann entropy to capture semantic similarities. In

addition to measuring the similarity between generated responses, Wang et al. (2024) proposed to judge the similarity between the target response and the generations. Duan et al. (2023) proposed Shifting Attention to Relevance (SAR), which focus on relevant components and assigns significance weights to tokens based on their contributions to the overall response. Unlike these carefully designed methods, Yaldiz et al. (2024) introduced a Learnable Response Scoring Function (LARS), which utilizes supervised data to capture complex token-probability dependencies. While effective, the above methods are computationally expensive. To alleviate these computational cost, Kossen et al. (2024) proposed Semantic Entropy Probes (SEPs) to approximate semantic entropy by leveraging hidden states from a single generation.

**Verbal confidence methods** Due to LLMs' strong language abilities and adherence to instructions, Verbal confidence methods are proposed. For instance, one may attach the question with a prompt like "Please respond and provide your confidence score ranging from 0 to 100.". Xiong et al. (2023) constructed a prompting, sampling, and aggregation framework to systematically evaluate various strategies and their integration. Groot & Valdenegro-Toro (2024) introduced FaR prompting, enhancing calibration by separating fact retrieval and reasoning. However, verbal confidence methods face significant challenges with over-confidence. Ni et al. (2024) found that LLMs cannot convey their uncertainties faithfully in natural language. Becker & Soatto (2024) found that combining language confidence and proxy model probability estimation can improve the estimation of uncertainty. Madhusudhan et al. (2024) noted LLMs' language perception accuracy often lags behind probability perception, especially in specific domains

**Internal state-based method** Internal state-based methods suggest that the activation of the target model can be analyzed to predict the model errors. Azaria & Mitchell (2023) proposed SAPLMA by training a classifier on the hidden layer activations of an LLM to assess statement truthfulness. Similarly, Liu et al. (2024) also introduced a supervised method by training a model on labeled datasets that analyze hidden layer activations and probability-related features. Focusing on the self-assessment capabilities of LLMs, Kadavath et al. (2022) trained models to explore the LLMs' ability to evaluate the accuracy of their responses through calibration on multiple-choice and true/false questions. Ji et al. (2024) employed a probing estimator to analyze the internal mechanisms of LLMs across various NLG tasks, assessing uncertainty before response generation. Additionally, some works introduced novel interventions to refine model performance during inference. Han et al. (2024) proposed to learn from past experience (LePe) method by leveraging historical performance records and fine-tuning instructions. Li et al. (2024a) presented Inference-Time Intervention (ITI) to adjust model activations selectively during inference across a limited number of attention heads, guided by a predefined set of directions.

**Consistency-based method** The consistency-based method is to evaluate the uncertainty of the large model through multiple generated answers. Recently, Li et al. (2024b) employed UQ sampling with perturbation and an aggregation module to quantify sampling uncertainty in text generation tasks. Pedapati et al. (2024) suggested reducing overconfidence by having LLMs justify answers and aggregate these to adjust confidence. Becker & Soatto (2024) proposed extracting semantic diversity and syntactic similarity from perturbed prompts, training a model on these features to estimate confidence. Yang et al. (2024) explored the stability of explanations generated by LLMs to estimate the model's confidence in its answers. Lin et al. (2023) discussed combining observed consistency and self-reflection to assess language model uncertainty

# 6 CONCLUSION

In this paper, We find that existing uncertainty estimation methods are often limited by the over-confidence and under-confidence inherent in LLMs, leading to inaccuracies in uncertainty estimation. To address these issues, we propose an external insight-driven correction approach which enables seamless integration with existing uncertainty estimation methods. We demonstrate that our method consistently outperforms existing approaches included in the Representative Baselines Set (RBS) and the Challenging Baselines Set (CBS), exhibiting significant improvements in both relative and absolute terms. Furthermore, we present comprehensive experimental evidence underscoring the robustness and generalizability of our approach across diverse data domains and target models.

ETHICS STATEMENT

In this study, we introduce a method for improving uncertainty estimation in the context of LLMs, which presents no immediate ethical concerns, but certain considerations must be addressed. Uncertainty estimation has significant potential to evaluate the reliability and safety of LLM outputs. However, this potential benefit comes with the risk that systematic mistakes in the uncertainty assessment could foster unfounded and misplaced confidence. Consequently, even re-calibrated uncertainty estimates should be interpreted cautiously, particularly in critical decision-making scenarios where the consequences of inaccuracies can be profound.

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
