# OpenReview forum: "Lookers-On See Most of the Game: An External Insight-Guided Method for Enhancing Uncertainty Estimation"
_ICLR.cc/2025/Conference — ICLR 2025 Conference Withdrawn Submission_

### Official Review · Reviewer_mpFG · 2024-11-01

**Soundness:** 2
**Presentation:** 2
**Contribution:** 2
**Rating:** 3
**Confidence:** 4

**Summary:**

The paper claims previous works on uncertainty estimation methods rely on model outputs, e.g., the logits or text, which cannot accurately measure the uncertainty due to over-confidence and under-confidence bias. To increase the accuracy of uncertainty estimation, the paper proposes to use a calibration method involving 1) selecting QA data from existing datasets with the same domain knowledge as the target evaluation task; 2) training a scorer that estimates the possibility of correctly answering the given question; 3) calibrating the traditional uncertainty score by adding the possibility of the model cannot correctly answer the given question.

**Strengths:**

1. The paper identifies the shortcomings of the existing uncertainty estimation methods -- relying on the outputs of the model confines uncertainty estimation to inherent biases of LLMs.
2. The paper proposes a calibration method that modifies the uncertainty scores by adding the probability that the model cannot correctly answer the given question. The calibration method can be applied to different existing uncertainty estimation methods.
3. The experimental results show that the accuracy of uncertainty estimation improves after calibration in TriviaQA and SciQA.

**Weaknesses:**

1. The method requires constructing a dataset with the same domain as the dataset used for evaluation.
    + The paper lacks details on the construction process.
    + How different data selection strategies influence the performance is unclear.
    + The construction needs the data to have the same domain as the target evaluation domain, which is difficult to scale/generalise to estimate uncertainty in real applications. The generalisation issue has been revealed by the performance decreases in Table 4.

2. The "external insight-guided method" and "lookers-on see most of the game" refer to using an external model to estimate the possibility of correctly answering the given question. The practical method is restricted by looking into a small constructed in-domain dataset and finetuning a RoBERTa-based model to estimate such probability. There could be different methods to discuss if the method focuses on estimating the probability of correctly answering the given question.

3. It is unclear how the method can be applied in non-question-answering tasks.

4. The paper discusses the preliminary in 2.5 pages, mixing with existing methods and the author's opinions. Most parts are not related to the proposed method, which confuses readers about the paper's contribution.

**Questions:**

See weakness.

---

### Official Review · Reviewer_LJ2y · 2024-11-01

**Soundness:** 2
**Presentation:** 3
**Contribution:** 2
**Rating:** 3
**Confidence:** 3

**Summary:**

This paper introduces an approach that trains an external lightweight model to mitigate inherent biases in LLMs, such as overconfidence and underconfidence. By benchmarking against established confidence estimation methods, the proposed model demonstrates enhanced performance and generalization capabilities across multiple models and datasets.

**Strengths:**

1. The proposed model shows substantial performance gains over baseline methods that do not incorporate training, demonstrating the effectiveness of the lightweight correction model in refining uncertainty scores.
2. Accurate uncertainty estimation is essential for the reliable deployment of LLMs in real-world applications. The proposed approach addresses this need by using an external lightweight model, which not only enhances reliability but is also designed to be resource-efficient.
3. The paper is well-organized, with a logical flow that makes it easy for readers to follow. Each section clearly builds on the last, enhancing comprehension. The paper thoroughly explains the training process, making the methodology accessible and reproducible.

**Weaknesses:**

1. **Missing Baselines:** This paper proposes a post-processing calibration method that requires task-specific training data, yet the baselines used for comparison are confidence elicitation techniques (pre-processing methods) that do not require training data, making the comparison potentially unfair. Established post-processing techniques like isotonic regression [1], Platt scaling [1], and recalibration with Bayesian neural networks [2] should be considered. Additionally, existing work has explored training an external calibrator specifically for LLMs [3]. A more balanced evaluation would fix the confidence estimation method and compare post-processing techniques that require training. The authors should cite relevant work, include applicable baselines, and justify any inapplicable baselines.
2. **Generalizability Concerns:** The authors claim generalizability across domains and models, but the approach’s intuition does not clearly support this. The corrector learns from an LLM’s biases on a specific task, so it’s unclear why it should generalize across models and tasks without similar biases. The current generalizability may stem from both tasks being relatively easy factoid QA, where models often exhibit comparable confidence biases. To substantiate generalizability, the authors should include more diverse tasks, such as those with varying difficulty (e.g., math), and test across models with different capabilities.
3. **Lack of Method Analysis:** The rationale behind this approach and its specific design remains unclear. The corrector functions as an evaluator trained on LLM data, yet it’s unclear:
- Why use an external model as the corrector rather than training the original model (or classifier based on the original model's hidden states)?
- How does this corrector complement the original uncertainty score? If given full weight, the corrector could solely determine the score—does improvement come from relying more on the corrector, or is there genuine complementarity?
- The paper claims the corrector mitigates the original model’s biases, but it’s unclear what specific biases (over-confidence, under-confidence) are present and addressed. An analysis of the corrector’s impact on these biases would clarify its effectiveness.
- The method’s success seems highly dependent on the corrector’s performance. Would substituting it with a strong LLM evaluator (e.g., GPT-4o) yield similar or superior results?
4. **Excessive Conceptual Background:** The paper devotes two pages (Section 2) to defining uncertainty concepts, which are widely understood and don’t directly influence the methodology or offer theoretical bounds. This lengthy section could distract readers and reduce space for more critical analysis and intuitive explanations of the method.

[1] Chuan Guo, et al. "On calibration of Modern Neural Networks".  https://arxiv.org/abs/1706.04599 \
[2] Juan Maronas, et al. "Calibration of Deep Probabilistic Models with Decoupled Bayesian Neural Networks"  https://arxiv.org/abs/1908.08972 \
[3]  Sabrina J. Mielke, et al. "Reducing conversational agents' overconfidence through linguistic calibration" https://arxiv.org/abs/2012.14983

**Questions:**

Questions are mentioned in weakness.

---

### Official Review · Reviewer_3nLV · 2024-11-03

**Soundness:** 2
**Presentation:** 3
**Contribution:** 2
**Rating:** 3
**Confidence:** 4

**Summary:**

This paper focuses on uncertainty estimation of LLMs. Motivated by existing methods, which might be limited by inherent biases of LLMs, the authors proposed to introduce an external corrector for refining uncertainty estimation, which is a Roberta-based regression model predicting a value added to the uncertainty score estimated by baselines. To train this corrector, they constructed a dataset consisting of questions, model responses, and correctness triplets. The corrector is trained to take the question and model response as inputs to predict the uncertainty.

**Strengths:**

1. Detailed theoretical analysis of preliminaries.
2. Experiment results confirm the effectiveness of the corrector.

**Weaknesses:**

1. Lacking comparisons of internal state-based methods (Azaria et al., [1]; Kapoor et al., [2]). Both approaches use a probing network to estimate model confidence from internal states. Since this work also involves training a corrector using additional constructed data, including these methods as comparisons would be valuable.
Besides, Liu et al. [3] generated their training set in a manner similar to this work and introduced a calibration mechanism to adjust logits, reducing the novelty of the proposed data construction process.
Additionally, Ulmer et al. [4] propose an external model to estimate the confidence of LLM outputs, closely resembling the proposed method, which also trains an external model for confidence prediction. This potentially diminishes the novelty of the external corrector.
2. The theoretical analysis in Section 2.3 states the limitations of using logits to estimate uncertainty in cases of over-confidence and under-confidence, but this is not applicable to other prior methods.

- [1] Azaria, A. and Mitchell, T., 2023. The internal state of an LLM knows when it's lying.
- [2] Kapoor, S., Gruver, N., Roberts, M., Collins, K., Pal, A., Bhatt, U., Weller, A., Dooley, S., Goldblum, M. and Wilson, A.G., 2024. Large Language Models Must Be Taught to Know What They Don't Know.
- [3] Liu, X., Khalifa, M. and Wang, L., 2024. LitCab: Lightweight Language Model Calibration over Short-and Long-form Responses.
- [4] Ulmer, D., Gubri, M., Lee, H., Yun, S. and Oh, S.J., 2024. Calibrating Large Language Models Using Their Generations Only.

**Questions:**

1. I’m somewhat confused by Step 3, which involves correcting uncertainty through a weighted combination of the uncertainty score from the corrector and other methods. However, there seems to be no specific design to train the corrector to ‘correct’ the uncertainty score; it is merely trained to align with the model’s performance. Why not use the uncertainty score directly from the corrector, as Ulmer et al., [4] did?
2. What are the selected values of $w^*$ in the experiments?

---

### Official Review · Reviewer_PxBm · 2024-11-04

**Soundness:** 1
**Presentation:** 3
**Contribution:** 1
**Rating:** 3
**Confidence:** 5

**Summary:**

The paper presents a training-based method to estimate uncertainty in LLMs. The method works by training a fully connected layer on top of RoBERTa on the accuracy data of the base LLM on the task. Then, a convex combination with another uncertainty estimation method is performed, using grid search to find the optimal hparm for this combination.

**Strengths:**

- The paper provides a nice mathematical formalization of overconfidence and underconfidence
- The paper is well-structured and well-written

**Weaknesses:**

- (Major) Lack of novelty:
	- The mathematical formalization of overconfidence and underconfidence is not very novel since it is a known issue in the literature. Formalizing it into a mathematical framework is a nice addition, but it's not clear its purpose since it is discarded soon after its definition and not used in the following sections.
	- The proposed Corrector is not novel since training an accuracy classifier is a known practice in the literature (e.g, [p(IK)](https://arxiv.org/abs/2207.05221), [Accuracy Probes](https://www.nature.com/articles/s41586-024-07421-0)) as well as training a more advanced classifier like [BERT](https://arxiv.org/abs/2402.11756)

- (Critical) The evaluation is quite weak, not thorough and with several issues:
	- The results proposed for RBS baseline methods seem suspiciously low and in contrast with known results in the literature. For example, on TriviaQA AUROC results of base methods are close to or below the 0.5 baselines in contrast with known results in the literature (https://aclanthology.org/2023.emnlp-demo.41.pdf , https://arxiv.org/abs/2406.15627 , https://www.nature.com/articles/s41586-024-07421-0). Please double-check your evaluation protocol.
	- Missing trivial baselines. The paper is missing several trivial single-sample logit-based baselines like Perplexity, Maximum Sequence Probability and Mean Token Entropy. Please take a look here and add these baselines (https://aclanthology.org/2023.emnlp-demo.41.pdf and https://arxiv.org/abs/2406.15627).
	- Missing more advanced baselines. The method proposed is based on training a classifier based on the correctness score of the base model on the task. However several methods in this same class (trained methods) are not evaluated and compared (e.g, [p(IK)](https://arxiv.org/abs/2207.05221), [Accuracy Probes](https://www.nature.com/articles/s41586-024-07421-0)). You should add these methods to your evaluation.
	- The results are presented in a weird way and no real comparison with other methods has been performed. Your method is presented as a method to boost a base score with a convex combination of your trained classifier. However, you did not compare with convex combinations of other baseline methods. The performance of the Corrector without the convex combination is not reported.
	- Your method has different requirements (training FLOPs, hparam tuning via grid search) that other simpler methods do not have. You did not take into consideration these different requirements when comparing methods and this analysis has been completely overlooked. You mention to optimize $w^*$ via grid search, but you provide no indication of the extensiveness of this search that is not performed on other methods. Your comparison should be fair and you should compare methods with equal requirements.
	- The results on CBS show minimal improvement w.r.t. to baseline methods on Llama-3 even though your method uses training. You should also report the confidence interval of your estimates to understand whether these improvements are meaningful. Performance on OPT-2.7B is misleading since the base model is not good enough and UE base methods rely on its logits.
	- Results are just on two datasets (TriviaQA and SciQA) and with two models (OPT- 2.7B, LLaMA-3-8B-Instruct). It's difficult to conclude something with this little evidence. It's not clear how the method would perform on different datasets and models. Furthermore, it's unclear whether the method scales with better base models compared to other approaches like p(True). This is important since UE methods rely on the model logits.

- (Minor) The paper contains several inaccuracies:
	- It's not clear the purpose of sections 2.1 . Framing uncertainty as a classification task (section 2.1) is the standard practice in the literature and you never clearly propose the definitions of relative uncertainty and absolute uncertainty.
	- I don't agree with equation 2. Equation 2 implies that uncertainty is the complement of the probability given by the model for a specific prompt. This is not always the case because some Uncertainty Estimation methods are not defined as probability (e.g., [INSIDE](https://openreview.net/forum?id=Zj12nzlQbz)).
	- I think equations 8 and 9 are wrong. Equation 9 should be $SE(x,\theta) = -\frac{1}{C} \sum_{i=1}^{C} P(c_i|x,\theta) \ln P(c_i|x,\theta)$ . Please check both of them
	- "internal state-based methods often lack stable theoretical frameworks". This is not true, there are some methods like [INSIDE](https://openreview.net/forum?id=Zj12nzlQbz) that have a "theoretical frameworks" that it's not based on output probabilities for trivial reasons. It's not clear what is your definition of "stable theoretical frameworks".
	- "showing their performance [internal state-based methods and verbalized methods] generally falling short of advanced logits-based methods". This is in contrast with results in the literature (e.g., [INSIDE](https://openreview.net/forum?id=Zj12nzlQbz))

- (Minor) The code proposed is missing several parts and files to understand how the evaluation has been performed, it's not usable and contains not-anonymized paths.

- (Minor) Your template is broken since it does not show lines, this is possibly due to compilation with latex errors.

**Questions:**

- I'm not sure about Equation 14, Jensen's inequality applies ≥  but you are using a stricter >. Can you explain why?
- "Based on the previous discussion, we can conclude that U(x) is overestimated due to the considerable contribution of 1 − p∗". Can you elaborate on this conclusion?
- Did you use p(TRUE) with few-shot examples?
- Can you report the raw score of your Corrector without the convex combination with the base UE methods?

---

### Official Review · Reviewer_bQtG · 2024-11-10

**Soundness:** 2
**Presentation:** 2
**Contribution:** 2
**Rating:** 3
**Confidence:** 4

**Summary:**

The paper proposes a method to address the challenge of improving uncertainty estimation in Large Language Models(LLMs) by training a "Corrector" model to refine the estimates made by existing methods. It provides theoretical and empirical evidence to demonstrate the tendency of LLMs to be over/under-confident, leading to over/under-estimating the uncertainty values. It creates a dataset comprising QA tasks to train the corrector model to align its outputs with that of the target model and proposes to correct the estimates by making it a weighted sum of the original and corrected estimates. The efficacy of the method is shown through experiments on QA tasks by reporting the AUROC and ECE scores, and ablations on the training/eval data domain and the target model.

**Strengths:**

1. The idea of utilizing an external model for uncertainty estimation in LLMs, instead of using internal representations like in existing logits-based or verbalized methods is novel and has the potential to be more unbiased.

**Weaknesses:**

1. The datasets/evaluation setup provides insufficient evidence to support the effectiveness of the corrector model. It seems fairly trivial that training a classification model on labeled examples would improve relative uncertainty scores, which are strongly correlated to the misclassification error for that domain and task. However, doing so worsens its objectivity in providing any complementary signal to the task performance. Moreover, the generalization of data domains shows negligible improvements across domains, which contradicts the claim that the corrector model or the method is general-purpose.

3. The writing of certain sections could be improved for clarity. For instance, the terminology of "representative" vs "challenging" is a confusing way of categorizing baseline methods in the context. Additionally, the claim that the relatively lower uncertainty scores of the former category are due to its "ease of implementation" isn't sufficiently justified. The experimental setup section also doesn't elaborate on the key information about the training data and uncertainty correction steps. It is also unclear from the results section how the proposed method addresses the over/under-confidence issue in LLMs over existing baselines, which makes the overall storyline less coherent.

**Questions:**

1. What dataset split was used as the validation set to perform the grid search over the weights in the correction step? What advantage does the weighted correction on baselines have over the external model directly predicting the estimate?
2. What is the scope of the proposed correction model? Is it meant to be a general-purpose model or a method to train for a specific domain and task?
3. In the theoretical work in section 2.3 - why does an absolute low/high uncertainty score imply that the estimate is over/under-estimated?

---

### Note · Authors · 2024-12-15

I have read and agree with the venue's withdrawal policy on behalf of myself and my co-authors.